The age of onset of substance use is related to the coping strategies to deal with treatment in men with substance use disorder

Capella Maria del Mar 1
Adan Ana aadan@ub.edu 1 2
1 Department of Clinical Psychology and Psychobiology, University of Barcelona , Barcelona , Spain
2 Institute of Neurosciences, University of Barcelona , Barcelona , Spain
Patton Bob
Electronic publication date: 2017 Aug 15
Publication date: 2017
Volume: 5
Electronic Location ID: e3660
Received 2017 Mar 13; Accepted 2017 Jul 17
Copyright: ©2017 Capella and Adan
Copyright year: 2017
Copyright holder: Capella and Adan
License: This is an open access article distributed under the terms of the Creative Commons Attribution License, which permits unrestricted use, distribution, reproduction and adaptation in any medium and for any purpose provided that it is properly attributed. For attribution, the original author(s), title, publication source (PeerJ) and either DOI or URL of the article must be cited.
License URL: https://creativecommons.org/licenses/by/4.0/

Keywords: Coping strategies, Addiction severity, Onset substance use, Treatment coping, Substance use disorders

Funding: Spanish Ministry of Science and Innovation PSI2009-12300 Spanish Ministry of Economy, Industry and Competitiveness PSI2012-32669 PSI2015-65026 This work was supported by grants from the Spanish Ministry of Science and Innovation PSI2009-12300, the Spanish Ministry of Economy, Industry and Competitiveness PSI2012-32669 and PSI2015-65026 (MINECO/FEDER/UE). The funders had no role in study design, data collection and analysis, decision to publish, or preparation of the manuscript.

==============================
Background

The age of onset of substance use (OSU) as well as the coping strategies (CS) influence both the development and the course of Substance Use Disorders (SUD). We aim to examine the differences in the CS applied to deal with treatment in men with SUD, considering whether the age of OSU had begun at age 16 or earlier (OSU ≤ 16) or at 17 years or later (OSU ≥ 17), as well as the associations of the CS with clinical variables were studied.

Methods

A total of 122 patients with at least three months of abstinence, 60 with OSU≤16 and 62 with OSU≥17, were evaluated through the Coping Strategies Inventory and clinical assessment tools.

Results

The OSU≤16 patients were younger and presented a worse clinical state. Compared to the norms, the SUD patients were less likely to use adaptive CS, although this was more remarkable for the OSU≤16 group. Furthermore, the OSU≤16 patients presented a CS pattern of higher Disengagement, with lesser use of Social Support and higher Problem Avoidance and Social Withdrawal. In the whole SUD sample, the severity of addiction, number of relapses and age of OSU (as a continuous variable) were related to maladaptive coping. Nevertheless, the cut-off age of OSU modulated these results.

Conclusions

The OSU≤16 was a risk factor for presenting greater clinical severity and a more dysfunctional CS profile to deal with treatment. Thus, the cut-off age considered has allowed us to differentiate SUD patients with more vulnerability to present worse clinical prognosis who may require specific prevention and rehabilitation strategies discussed throughout this work.

Introduction

Substance Use Disorders (SUD) are considered a public health issue since they have severe personal and community consequences, as well as a high worldwide prevalence (United Nations Office on Drugs and Crime (UNODC) 2015). Despite welfare programs, patients have a high variability of response to interventions (Kampman et al., 2007) and high rates of relapse (Suijkerbuijk et al., 2015; Witkiewitz & Marlatt, 2004). Moreover, the age of onset of substance use (OSU) usually occurs at very early ages and it has been established as a strong predictor to future SUD (Woodcock, Lundahl & Stoltman, 2015) and linked both to worse clinical course and cognitive functioning (Capella, Benaiges & Adan, 2015; Eddie, Epstein & Cohn, 2015; Hammond, Mayes & Potenza, 2014; Kendler et al., 2013) and greater brain alterations (Elofson, Gongvatana & Carey, 2013). Thus, it is necessary to further study the risk factors and course severity in SUD, such as age of OSU, with the aim of improving prevention and treatment strategies according to their characteristics, to make them more effective.

In this line, we have attempted to clarify the reasons that lead people to consume substances. While the age of OSU is often linked to social goals and to its positive reinforcement, once SUD consumption has developed it is maintained in order to reduce the stress-based negative affect (Blevins et al., 2014; Dermody, Cheong & Manuck, 2013). Although there are multiple factors that influence the clinical prognosis of SUD patients, certain psychological characteristics are essential. Among these, we can mention the coping strategies (CS) used to face adversity since it has been shown that they influence both the development and course of SUD and its treatment outcome (Marquez-Arrico, Benaiges & Adan, 2015; Walker & Stephens, 2014).

Lazarus & Folkman (1984) define CS as cognitive and behavioral responses aimed at managing internal or external demands. They have established two major ways of coping: engagement, aimed at dealing with the stressors or their related emotions, and generally considerate adaptive; and disengagement, targeted to avoid the stressful situations or their related emotions, and mainly regarded as maladaptive (Carver & Connor-Smith, 2010; Skinner et al., 2003; Tobin et al., 1989). People are likely to show a relatively stable disposition towards the habitual use of certain CS to diverse stressors, which to some extent would vary depending on situation-specific variables in a concrete coping episode or stressor (Bauer et al., 2016; Bouchard, Guillemette & Landry-Léger, 2004; Lazarus & Folkman, 1984) and age (Mauro et al., 2015; Woodhead et al., 2014).

Substance consumption is considered as a type of coping behavior to avoid stress, focused on emotions and directed to temporarily alleviate the negative affect that certain stressors generate, although in the long term it will trigger more severe problems, such as the need to consume again (Bavojdan, Towhidi & Rahmati, 2011; Buckner et al., 2015; Hruska et al., 2011). The avoidance-based coping style has been established as a risk factor with a poor prognosis for the initiation and maintenance of SUD. In contrast, problem-focused coping is considered a protective factor against consumption (Blevins et al., 2014; Coriale et al., 2012; Dermody, Cheong & Manuck, 2013; Marquez-Arrico, Benaiges & Adan, 2015; Walker & Stephens, 2014; Woodhead et al., 2014) and is associated to better mental health (Bavojdan, Towhidi & Rahmati, 2011; Nyamathi et al., 2010). Similarly, social support is a CS with a protective effect against stress (Hyman et al., 2009) and the development of depression (Aarts et al., 2015), having been observed that it correlates negatively with the relapse rate of SUD patients (Chauchard, Septfons & Chabrol, 2013; Dolan et al., 2013; Hägele et al., 2014). However, recent results are controversial (Nyamathi et al., 2010) and need to be contrasted.

Adolescence is a risk stage for the onset of several psychiatric disorders, such as those related with substance use (Hägele et al., 2014; Kirst et al., 2014). Since early OSU is associated to future SUD development (Woodcock, Lundahl & Stoltman, 2015) and to more severe characteristics (Capella, Benaiges & Adan, 2015; Eddie, Epstein & Cohn, 2015; Hammond, Mayes & Potenza, 2014; Kendler et al., 2013), studying the CS of addicts and their clinical implications, considering the age of OSU, is a research area of undoubted clinical interest.

Based on characteristics of brain ontogeny (Kunert, Derichs & Irle, 1996; Lambe, Krimer & Goldman-Rakic, 2000; Shaw et al., 2006; Sundram, 2006), previous studies have found that patients who begin consumption at age 16 or earlier, when compared to those with onset at age 17 or later, have a lower premorbid intelligence quotient (Capella, Benaiges & Adan, 2015; Pope et al., 2003), worse neuropsychological performance (Ehrenreic et al., 1999; Jockers-Scherübl et al., 2007) and less cerebral and gray matter volume (Wilson et al., 2000). However, considering this cut-off age, no previous studies have provided data about the possible differences in CS patterns and their relationship with clinical variables.

Our paper has two aims. The first is to assess the differences in the CS profile to deal with the treatment of men diagnosed with SUD, depending on whether they initiated substance use at age 16 or earlier (OSU≤16) or at age 17 or later (OSU≥17), as well as in relation to normative data. The second is to explore the relationships among CS- and SUD-related clinical variables.

Method

Study design and participants

We enrolled 122 patients under SUD treatment in different healthcare resources (ambulatory drug use treating or residential in therapeutic community) in a cross-sectional study design. All were male, given the high prevalence of this gender in SUD (UNODC, 2015) and to avoid biasing the results due to sex differences (Woodhead et al., 2014). Participants were derived from treatment centers after selection according our inclusion/exclusion criteria. In a first evaluation session we confirmed the diagnosis by a researcher (trained clinical psychology postgraduate) responsible of clinical assessment. In a second session, the coping strategies assessment tool along with other tests not presented in this manuscript were administered. After collecting data, they were assigned to two groups according to age of OSU: one for OSU at age 16 or earlier (OSU≤16; n = 60), and one for OSU at age 17 or later (OSU≥17; n = 62). The consideration of this age cut-off was based on the neurodevelopmental characteristics (Kunert, Derichs & Irle, 1996; Lambe, Krimer & Goldman-Rakic, 2000; Shaw et al., 2006; Sundram, 2006), as well as the differences in cognitive performance found in previous studies (Capella, Benaiges & Adan, 2015; Ehrenreic et al., 1999; Jockers-Scherübl et al., 2007; Pope et al., 2003).

The inclusion criteria were: (1) current or past diagnosis of SUD confirmed by a diagnostic interview according to the criteria in the Diagnostic and Statistical Manual of Mental Disorders, Fourth Edition Text Revised (DSM-IV-TR; American Psychiatric Association, 2000); (2) with abstinence for at least three months at the time of the study (excluding caffeine or nicotine consumption), confirmed by urinalysis, to ensure the overcoming of withdrawal symptoms and minimum adherence to treatment; (3) age between 18–55 years. The exclusion criteria were: (1) presence of mental retardation or pervasive developmental disorder, history of traumatic brain injury, neurological injury or any other medical problem which could interfere in the assessment; (2) presence of a comorbid axis I mental disorder, such mood or affective disorders, confirmed by a diagnostic interview according to DSM-IV-TR criteria.

All patients provided written informed consent and were not compensated for their participation. The ethic committee of the University of Barcelona approved this study (IRB00003099), which meets the ethical principles of the declaration of Helsinki (World Medical Association, 2013). This study was part of a larger project on clinical characteristics, neuropsychological functioning, personality traits and circadian rhythmicity in SUD and Dual Diagnosis patients.

Clinical and sociodemographic measures

Through a structured interview designed specifically for our study and the Structural Clinical Interview for the DSM-IV Axis I Disorders (SCID-I; First et al., 1999), we collected sociodemographic (age, marital, educational and economic status) and clinical data (presence of psychiatric pathology and substance use family history, suicidal attempts, past treatment for SUD, consumption pattern, type of drugs used, age of OSU, duration of drug use, residential or ambulatory treatment, medication, abstinence periods and relapses). This information was confirmed with the medical history of the centers’ databases and with the patients’ treating psychiatrist.

The Clinical Global Impression questionnaire (CGI; Guy, 1976) was administered as a subjective measure of clinical severity. Furthermore, severity of SUD was assessed using the Drug Abuse Screening Test (DAST-20; Skinner, 1982) through its Spanish version (Gálvez & Fernández, 2010), which provides a total score from 0 to 20 (0 no addiction, 1–5 low, 6–10 intermediate, 11–15 substantial, and 16–20 severe).

Coping strategies assessment

CS were assessed by means of the Spanish version (Cano-García, Rodríguez-Franco & Martínez, 2007) of the Coping Strategies Inventory (CSI; Tobin et al., 1989). Patients assessed the frequency with which they had used the strategies described to deal with their SUD treatment. The CSI is composed by 41 items with 5-point Likert scale answers, of which 40 configure the primary scales with one additional item on self-perceived coping ability. The CSI has a hierarchical structure composed by eight primary, four secondary and two tertiary scales. The primary scales are: Problem Solving, Cognitive Restructuring, Social Support, Express Emotions, Problem Avoidance, Wishful Thinking, Social Withdrawal, and Self-Criticism. The secondary scales are: Problem Focused Engagement (composed by Problem Solving and Cognitive Restructuring), Emotion Focused Engagement (Social Support and Express Emotions), Problem Focused Disengagement (Problem Avoidance and Wishful Thinking) and Emotion Focused Disengagement (Social Withdrawal and Self Criticism). The tertiary scales of the CSI are Engagement (Problem and Emotion Focused Engagement) and Disengagement (Problem and Emotion Focused Disengagement).

Statistical analyses

Descriptive statistics and frequencies were calculated to describe the total study sample. Differences in sociodemographic and clinical variables between groups were explored with the Mann–Whitney U test (U) or with the Chi Square test (χ2) for categorical variables. The Student’s t-test (t) was used when the quantitative data fulfilled the necessary conditions, and the U test was used instead when those conditions were not met. Factorial analysis of eight factors for CSI was performed with a normalized varimax rotation employed to achieve factor simplicity. Furthermore, internal consistency for the primary scales was calculated with the Cronbach’s alpha coefficients. Three multivariate analyses of covariance (MANCOVA) were performed considering the primary, secondary and tertiary CSI scales. Age was considered as covariate, since it could be a confounding factor (Mauro et al., 2015; Woodhead et al., 2014). An additional analysis was carried out considering the treatment regimen (residential or ambulatory), to assess whether this was an indicator of differences in the CS profile to deal with treatment related to the recruitment of patients and not related to the age of OSU. The Bonferroni test was applied in all analyses to reduce the occurrence of a type I error. The effect size was calculated with the partial Eta squared (ηp2), assuming a value of 0.01 as low, of 0.04 as moderate and of 0.1 as high (Huberty, 2002). The data were compared to the Spanish norms, only available for the primary scales (Cano-García, Rodríguez-Franco & Martínez, 2007), using percentiles.

The relationships between CS and SUD clinical variables were studied in two steps, both for the total sample and for each group. First, we carried out correlational analyses between CS and clinical data; then, the significant results were introduced in a multiple stepwise regression analyses with CS as dependent variables.

Data were analyzed using the Statistical Package for the Social Sciences (SPSS; version 15.0), considering bilateral statistical significance with an established type I error at 5% (p < .05).

Results

Differences in sociodemographic and clinical data

The total sample was aged 20 to 55 (M = 35.97, SD = 8.31) and most of the patients had completed the Spanish compulsory education (from 6 to 16 years). Regarding sociodemographic variables, the only observed difference between groups was the lower mean age of the OSU≤16 group (p < .001), while both groups did not differ in years of education, marital and economic status. The analyses of the clinical variables provided no significant differences between groups regarding relatives with other psychiatric disorders than SUD and number of suicidal attempts. Instead, in the OSU≤16 group it was more frequent to have relatives with SUD (p = .025). See Table 1.

Table 1 Descriptive statistics (frequencies or mean and standard error) of the sociodemographic and clinical data, for the total sample and groups, and the statistical contrasts carried out.

	Total sample (N = 122)	OSU≤16 (N = 60)	OSU≥17 (N = 62)	Statistical contrasts	
Sociodemographic data					
Age (yr)	35.97 (0.75)	33.60 (1.09)	38.25 (0.96)	U= 1,159.50***	
Years of education	10.36 (0.23)	10.25 (0.33)	10.48 (0.32)	t(120) =  − 0.476	
Marital status				χ2(1) = 5.015	
Single	51.6%	51.7%	51.6%		
Separate/Divorced	22.1%	21.6%	22.5%		
Married	14.8%	13.3%	16.1%		
Stable partner	11.5%	13.3%	9.7%		
Economic status				χ2(1) = 4.305	
Unemployed	29.5%	30%	29%		
Active	23.8%	16.7%	30.6%		
No income	19.7%	23.3%	16.1%		
Disability pension	16.4%	20%	12.9%		
Sick leave	10.7%	10%	11.3%		
Clinical data					
Relatives with SUD	26.2%	25%	9.7%	χ2(1) = 5.024*	
Relatives with others psychiatric disorder	26.1%	23.3%	29%	χ2(1) = 0.512	
Number of suicidal attempts	0.23 (0.66)	0.30 (0.10)	0.16 (0.06)	t(120) = 1.144	
Notes.

OSU≤16 Onset of substance use at age 16 or earlier

OSU≥17 Onset of substance use at age 17 or later

yr years

SUD Substance Use Disorder

* p < .05.

*** p < .001.

With respect to SUD data, the OSU≤16 group had higher rates of polyconsumption (p = .030), and patients in residential rather than ambulatory treatment (p = .013), lower age of OSU (p < .001) and longer duration of drug use (p = .016). Furthermore, the groups showed differences in the type of substances used. In the OSU≤16 group, there were higher rates of cannabis (p < .001), and hallucinogens consumption (p = .042), while in the OSU≥17 group there are higher rates of cocaine consumption (p = .046). In the overall sample, as well as in both groups, the substances more frequently used were cocaine, alcohol and cannabis. No differences between groups were found in the other SUD clinical characteristics studied (see Table 2).

Table 2 Descriptive statistics (frequencies or mean and standard error) of the data related to SUD, for the total sample and groups, and the statistical contrasts carried out.

SUD clinical characteristics	Total sample (N = 122)	OSU≤16 (N = 60)	OSU≥17 (N = 62)	Statistical contrasts	
Consumption pattern					
One substance	23.8%	18.3%	29%	U = 1,661	
Two substances	34.4%	30%	38.7%	U = 1,698	
Polydrug use	41.8%	51.7%	32.3%	U = 1,499*	
Substances useda					
Cocaine	87.7%	81.7%	93.5%	χ2(1) = 3.992*	
Alcohol	72.1%	76.7%	67.7%	χ2(1) = 1.208	
Cannabis	43.4%	60%	27.4%	χ2(1) = 13.173***	
Hallucinogens	16.4%	23.3%	9.7%	χ2(1) = 4.149*	
Opioids	14.8%	16.7%	12.9%	χ2(1) = 0.343	
Sedatives	4.1%	5%	3.2%	χ2(1) = 0.244	
Age of OSU (yr)	19.16 (0.59)	14.98 (0.15)	23.20 (0.88)	U = 17.50***	
Duration of drug use (yr)	16.13 (0.77)	17.98 (1.07)	14.33 (1.06)	t(120) = 2.432*	
Typology of treatment regimen				χ2(1) = 6.124*	
Residential	62.3%	73.3%	51.6%		
Ambulatory	37.7%	26.7%	48.4%		
Daily number of medication	0.51 (0.08)	0.65 (0.14)	0.36 (0.08)	t(120) = 1.813	
Months of abstinence	8.14 (0.47)	7.60 (0.59)	8.66 (0.72)	t(120) =  − 1.127	
Past treatment for SUD				χ2(1) = 0.085	
Yes	50.5%	51.9%	48.9%		
Number of relapses					
None	57%	56.7%	57.4%	U = 1,856	
One	16.5%	13.3%	19.7%	U = 1,718	
Two	10.7%	15%	6.6%	U = 1,701	
Three or more	15.7%	15%	16.4%	U = 1,825.50	
Drug Abuse Screening Test (DAST-20)	12.15 (0.47)	12.33 (0.57)	11.94 (0.78)	t(1) = 0.418	
Clinical Global Impression (CGI)	2.45 (0.11)	2.58 (0.17)	2.33 (0.15)	t(1) = 1.069	
Notes.

SUD Substance Use Disorder

OSU≤16 Onset of substance use at age 16 or earlier

OSU≥17 Onset of substance use at age 17 or later

OSU Onset of substance use

yr years

DAST-20 Drug Abuse Screening Test

CGI Clinical Global Impression

a Percentages will not equal 100 as each participant may take more than one substance of abuse.

* p < .05.

*** p < .001.

Coping Strategies Inventory comparisons

The eight factors obtained from factorial analysis of the CSI explained the 61.95% of variance (F1 = 9.73, F2 = 9.55, F3 = 8.61; F4 = 8.35; F5 = 7.93; F6 = 7.16, F7 = 5.34 and F8 = 5.29). The set of items of the Problem Solving and Wishful Thinking scales are the most clearly associated with one single factor, with loadings superior to .43. Regarding the scales of Self-Criticism, Express Emotions and Cognitive Restructuring scales, only four of the five items selected converge in the factors, with values higher to .44. Finally, Social Support, Problem Avoidance and Social Withdrawal are the scales that correspond less to the items of the originals.

Cronbach’s alpha coefficients of internal consistency for the primary scales were all adequate for the total sample studied: Problem Solving (0.79), Cognitive Restructuring (0.71), Social Support (0.72), Express Emotions (0.76), Problem Avoidance (0.70), Wishful Thinking (0.77), Social Withdrawal (0.74) and Self-Criticism (0.77).

The comparisons among percentile scores in the CSI primary scales according to the Spanish normative data (see Fig. 1) showed lower scores in the overall sample (<40 percentile) in Problem Solving, and higher scores (>60 percentile) in Express Emotions, Wishful Thinking, Social Withdrawal and Self-Criticism. When considering groups, higher scores in Wishful Thinking, Social Withdrawal and Self-Criticism were observed in both. On the other hand, the OSU ≥17 group had higher scores in Social Support and lower scores in Problem Avoidance.

Figure 1 Mean of percentile scores for the Coping Strategies Inventory.

OSU≤16: Onset of substance use at age 16 or earlier; OSU≥17: Onset of substance use at age 17 or later.

The MANCOVA analyses showed several significant differences between groups for the CSI scales (see Table 3). Results in the primary scales indicated that the OSU≤16 group had lower scores in Social Support (p = .019), and higher scores both in Problem Avoidance (p = .037), and in Social Withdrawal (p = .049). No differences between groups were found in the other primary and secondary scales. Regarding tertiary scales, the groups were similar in their use of Disengagement strategies, but the OSU≤16 group showed lower scores in Engagement strategies (p = .038).

Table 3 Means and standard deviations for the total sample and for each group, and results of the MANCOVA analyses for both groups considering age as a covariate for the Coping Strategies Inventory (CSI).

CSI	Total sample (N = 122)	OSU≤16 (n = 60)	OSU≥17 (n = 62)	F	Effect size	
Primary subscales						
Problem solving	13.31 (4.76)	12.83 (4.69)	13.77 (4.82)	0.835	0.007	
Cognitive restructuring	10.44 (5.04)	9.72 (5.05)	11.16 (4.97)	2.705	0.022	
Social support	11.50 (5.03)	10.50 (5.35)	12.48 (4.52)	5.628*	0.045	
Express emotions	10.37 (5.08)	10.70 (5.25)	10.05 (4.94)	0.891	0.007	
Problem avoidance	5.77 (4.64)	6.85 (5.32)	4.76 (3.63)	4.471*	0.036	
Wishful thinking	14.69 (4.81)	14.60 (4.67)	14.77 (4.99)	0.004	0.001	
Social withdrawal	9.68 (5.30)	10.77 (5.18)	8.63 (5.25)	3.959*	0.029	
Self-criticism	13.57 (5.09)	13.48 (5.29)	13.66 (4.95)	0.081	0.011	
Self-perceived capacity	2.70 (1.33)	2.50 (1.42)	2.90 (1.21)	1.245	0.010	
Secondary subscales						
Problem focused engagement	23.78 (8.08)	22.60 (8.29)	24.91 (7.77)	2.353	0.019	
Emotion focused engagement	21.79 (8.10)	21.10 (8.71)	22.45 (7.48)	0.772	0.006	
Problem focused disengagement	20.53 (5.92)	21.45 (6.21)	19.62 (5.52)	2.562	0.021	
Emotion focused disengagement	23.48 (8.64)	24.25 (8.28)	22.72 (8.98)	0.423	0.004	
Tertiary subscales						
Engagement	45.59 (14.26)	43.70 (14.98)	47.42 (13.39)	1.944	0.016	
Disengagement	43.61 (10.92)	45.87 (11.16)	41.44 (10.31)	4.417*	0.036	
Notes.

OSU≤16 Onset of substance use at age 16 or earlier

OSU≥17 Onset of substance use at age 17 or later

* p < .05.

In the additional analysis carried out considering the treatment regimen (residential or ambulatory) as a group variable, no significant differences between groups were found regarding any primary, secondary or tertiary CSI scales (p > .241; in all cases), neither considering age and age of OSU as covariates (p > .240; in all cases).

Relations among SUD, clinical characteristics and coping strategies

First, correlations between CS and clinical data, and between CS and SUD clinical characteristics, indicated that only age of OSU (considered as a continuous variable), duration of drug use, number of relapses and DAST-20 were associated with some of the CS: Wishful Thinking, Social Withdrawal, Problem Focused Disengagement, Emotion Focused Disengagement and Disengagement.

Second, the above variables were introduced in regression analyses with the different CS as dependent variables (see Table 4). In the total sample, the DAST-20 explained 11% of the variance of Problem Focused Disengagement (p = .002), 13% of Social Withdrawal (p = .001), 17% of Wishful Thinking (p < .001) and, together with age of OSU and number of relapses, 23% of the variance of Emotion Focused Disengagement (p < .001). Age of OSU and number of relapses accounted for 15% of the variance of Disengagement (p = .002).

Table 4 Multiple linear regression for the Coping Estrategies.

Multiple linear regression for the Coping Strategies Inventory (CSI) considering as independent variables the sociodemographic and clinical data that showed significant correlations, for the total sample (N = 122) and for the OSU≥16 (N = 60) and OSU≥17 (N = 62) groups.

CSI	Adjusted R2	F	IVa	β Standardized	p values	
Total sample						
Wishful thinking	0.167	15.279***	DAST-20	0.423	.0001	
Social withdrawal	0.133	11.900**	DAST-20	0.381	.001	
Problem focused disengagement	0.113	10.022**	DAST-20	0.166	.002	
Emotion focused disengagement	0.232	8.157***	Age of OSU (yr)	−0.276	.010	
			Number of relapses	0.220	.047	
			DAST-20	0.302	.007	
Disengagement	0.146	7.072**	Age of OSU (yr)	−0.269	.017	
			Number of relapses	0.300	.008	
OSU≤16						
Wishful thinking	0.187	9.732**	DAST-20	0.456	.003	
Social withdrawal	0.265	7.835**	Number of relapses	0.322	.029	
			DAST-20	0.387	.010	
Problem focused disengagementb						
Emotion focused disengagement	0.218	6.307**	Number of relapses	0.315	.038	
			DAST-20	0.342	.025	
Disengagement	0.120	6.159*	Number of relapses	0.378	.018	
OSU≥17						
Wishful thinking	0.131	5.827*	DAST-20	0.398	.022	
Social withdrawal	0.105	4.740*	DAST-20	0.364	.037	
Problem focused disengagement	0.171	7.577*	DAST-20	0.443	.010	
Emotion focused disengagement	0.115	5.138*	DAST-20	0.377	.031	
Disengagementb						
Notes.

OSU≤16 Onset of substance use at age 16 or earlier

OSU ≥17 Onset of substance use at age 17 or later

IV Independent Variables

DAST-20 Drug Abuse Screening Test

OSU Onset of Substance Use

yr years

a Only significant variables are presented that comprise each explicative model. In all cases, the Tolerance values were higher than 0.91 and the Variance Inflation Factor values lower than 1.09.

b Any explicative model was significant.

* p < .05.

** p < .01.

*** p < .001.

Considering the OSU≤16 group, the regression analysis indicated that the model was significant in only four of the CS. The DAST-20 explained 18% of the variance of Wishful Thinking (p = .003). The DAST-20 and the number of relapses were significant for Social Withdrawal (p = .001) and Emotion Focused Disengagement (p = .004), explaining 27% and 22% of the variance, respectively. Number of relapses explained 12% of the variance of Disengagement (p = .018).

Moreover, only four significant regression models were observed in the OSU≥17 group. In this case, the DAST-20 were significant for Wishful Thinking (p = .022), Social Withdrawal (p = .037), Problem Focused Disengagement (p = .010) and Emotion Focused Disengagement (p = .031), accounting for 11%, 12%, 13% and 17% of the variance, respectively.

Discussion

To our knowledge, this is the first study that aims to elucidate the possible existence of differences in CS related to treatment in men with SUD, depending on whether their substance use began at age 16 or earlier, or at age 17 or later. In addition, we have also assessed the influence of clinical variables related to the SUD regarding the CS pattern.

Sociodemographic and clinical differences between groups

In relation to the sociodemographic data, both groups differ only in age, the patients in the OSU≤16 group being the youngest. We decided to control the possible effect of this variable on the coping results, given that with the passing of years people may develop some variation in the CS they use (Mauro et al., 2015; Woodhead et al., 2014). The substances more frequently consumed in both groups were cocaine, alcohol and cannabis. However, the patients in the OSU≤16 group had a higher frequency of cannabis and hallucinogens consumption, whereas cocaine was the most frequently consumed drug in the OSU≥17 group. These differences may reflect the social preference for a certain substance at the time the patients were developing their SUD (European Monitoring Centre for Drugs and Drug Addiction, 2016).

When the cut-off age of OSU was considered, the clinical characteristics of both groups fall in line with the only data currently available (Capella, Benaiges & Adan, 2015), that is, the OSU≤16 patients present a more severe clinical and SUD pattern, characterized by more substance consumption, greater duration of drug use and the need for a more intensive treatment (residential instead of ambulatory) to achieve abstinence. Other works have analyzed the age of OSU, although without establishing a cut-off point, and have obtained similar results (Eddie, Epstein & Cohn, 2015; Hammond, Mayes & Potenza, 2014; Kendler et al., 2013). Moreover, the higher presence of a family history of SUD in the OSU≤16 group supports the findings on the genetic predisposition and early environmental exposures that precede the onset of drug use in early consumers (Hägele et al., 2014; Hammond, Mayes & Potenza, 2014). These data should be further explored in future research taking into account the possible mediation of the main substance consumed, since this family factor has not yet been found in cannabis consumers (Pope et al., 2003). Finally, it is interesting to note that the CS profile of the patients was independent of the type of treatment regimen in agreement with a previous study (Adan, Antúnez & Navarro, 2017).

Coping strategies

The SUD patients show a tendency to use emotion in order to cope with the stress generated by their treatment. There is a predominance to use maladaptive CS such as avoiding any contact with those persons related to their stressful experience (Social Withdrawal), as well as self-blame for the occurrence of the stressful situation or its improper management (Self-Criticism). When considering the most adaptive pole, the predominant CS is the release of emotions that occur in the process of stress (Express Emotions). When focusing on addressing the problem, the tendency was to do it in an inappropriate way by thinking about non-stressful alternative realities (Wishful Thinking), with a low propensity to active resolution of their difficulties (Problem Solving). This is consistent with the results of previous studies that have found a tendency to maladaptive coping, with lesser use of active coping problem-solving strategies in drug consumers (Marquez-Arrico, Benaiges & Adan, 2015), which could be at the basis of the onset (Walker & Stephens, 2014; Woodhead et al., 2014) and maintenance of the SUD (Coriale et al., 2012). The pattern described is observed in both groups, although the OSU ≥17 group presents a more adaptive coping profile.

When compared to the OSU≥17, the OSU≤16 group is characterized by a CS pattern of greater Disengagement (lesser use of Social Support; greater Problem Avoidance and Social Withdrawal). Thus, it seems that the patients with an early OSU present fewer interpersonal skills, showing a tendency to withdraw from their social environment and to avoid contact with other people in order to express their emotions. In this sense, we find that a higher rate of substance and medication use in adults is related to a lesser early childhood social competence (Jones, Greenberg & Crowley, 2015). Furthermore, in SUD patients under treatment high rates of cooperation (Andó et al., 2012) and stronger social support (Chauchard, Septfons & Chabrol, 2013; Dolan et al., 2013) have been related to longer abstinence periods, while social withdrawal is higher when the age of OSU is younger (Marquez-Arrico, Benaiges & Adan, 2015). Moreover, avoidance-based coping has been consistently identified as a moderator of SUD (Bavojdan, Towhidi & Rahmati, 2011; Hruska et al., 2011), and this could be a key factor in the tension-reduced-based models, where emotional suffering is relieved through avoidance of emotional distress (Buckner et al., 2015).

Influence on coping strategies of SUD clinical characteristics

The relations observed among the clinical variables and maladaptive coping are modulated by the age of OSU and, specifically, by the cut-off age studied in this work. In SUD patients, a higher severity of addiction is related to a more dysfunctional coping pattern when they face treatment (higher scores in Wishful Thinking, Social Withdrawal, Problem Focused Disengagement and Emotion Focused Disengagement). Moreover, a higher number of relapses is related to a more frequent use of a coping style based on emotional disengagement (Emotion Focused Disengagement and Disengagement strategies), which is explained by the pattern of the OSU≤16 group. Thus, those patients with a more severe SUD tend to use maladaptive CS (Blevins et al., 2014; Marquez-Arrico, Benaiges & Adan, 2015; Nyamathi et al., 2010), and this is related to a higher number of relapses (Chauchard, Septfons & Chabrol, 2013; Dolan et al., 2013), even the more so as the age of OSU decreases.

There are some limitations in our study that should be mentioned. Almost half of the patients were polyconsumers, which was an impediment to assess separately the effect of each type of substance on CS. However, their possible effect was relatively controlled since the groups had consumed the same main substances (cocaine, alcohol and cannabis). The inclusion of only men in the sample, knowing the influence of gender in CS (Bouchard, Guillemette & Landry-Léger, 2004; Woodhead et al., 2014), limits the generalization of our results. The wide range of age in the sample may have also contributed to a type-II error. We have analyzed cross-sectional data, which does not allow establishing causal or sequential relations among variables, or if the coping strategies currently used by patients are the same as when they started the substance use or even the treatment. Moreover, the level of perceived stress and affect or mood of participants were not assessed, which could help explain their CS pattern and clinical traits (Aarts et al., 2015; Hyman et al., 2009). While the data obtained in the CSI indicated an adequate internal consistency for the primary scales, the factorial analysis with the eight factor structure model of the Spanish version (Cano-García, Rodríguez-Franco & Martínez, 2007) was only partially fulfilled, although it is widely agreed and validated. Finally, the low explanatory power obtained in the regression analyses warns us to interpret the results with caution. Future works should include larger patient samples of both men and women, differentiating the main substance consumed, with a longitudinal design that could contribute to clarify the attitudinal or situational character of coping and which variables determine it in order to obtain a better knowledge of its role in later outcomes.

Our results may have clinical interest. We find that SUD patients deal with treatment applying a dysfunctional emotion-based coping style, a style also present at the onset, maintenance and severity of the disorder (Bavojdan, Towhidi & Rahmati, 2011; Blevins et al., 2014; Buckner et al., 2015; Dermody, Cheong & Manuck, 2013; Hruska et al., 2011). This suggests that those interventions aimed at developing CS which may prevent consumption, such as problem-focused coping o social support, could be more effective both in addiction prevention and treatment programs. The assessment of cognitive skills seems another key factor to be considered, given that a low neuropsychological performance is related both to the development of SUD in adulthood (Pechtel, Woodman & Lyons-Ruth, 2012) and to the response to treatment (Kiluk, Nich & Carroll, 2011). In this regard, it has been shown that, in the rehabilitation of SUD patients, higher cognitive abilities are associated with greater improvement in the quality of the coping skills acquired, which in turn is indirectly associated with treatment benefits. This could be significantly relevant for patients with OSU≤16, since they present a worse coping style, as shown in the present study, and a lower cognitive performance, as shown in other works (Capella, Benaiges & Adan, 2015; Ehrenreic et al., 1999; Jockers-Scherübl et al., 2007; Pope et al., 2003). Further research is required to shed light on this issue.

Conclusions

A main priority in public health should be to target populations at risk of developing SUD or with a worse clinical prognosis, in order to design more specific intervention programs. In this sense, the cut-off age considered in our study is a contribution, since the OSU≤16 patients exhibited more vulnerability to present both higher clinical severity and frequency to use of a dysfunctional CS profile to cope with treatment, which were related to the severity of the addiction and relapses. Further studies are needed to explore the possible benefits of improving adaptive coping in these patients for better treatment outcomes, as well as for prevention programs of SUD.

Supplemental Information

Data S1 Supplemental data

Click here for additional data file.

Data S2 Factorial analysis CSI

Click here for additional data file.

We thank the Man Project Foundation in Catalonia, the ATRA Association and the Mental Health and Addictions Division of the Mataró Hospital for providing the patients in the sample.

Additional Information and Declarations

Competing Interests

Author Contributions

Human Ethics

Data Availability

The authors declare there are no competing interests.

Maria del Mar Capella performed the experiments, analyzed the data, contributed reagents/materials/analysis tools, wrote the paper, prepared figures and/or tables, reviewed drafts of the paper.

Ana Adan conceived and designed the experiments, contributed reagents/materials/analysis tools, wrote the paper, prepared figures and/or tables, reviewed drafts of the paper, conceived the original idea for the study, sought funding, and wrote the protocol.

The following information was supplied relating to ethical approvals (i.e., approving body and any reference numbers):

The ethics committee of the University of Barcelona granted approval to carry out the study.

The following information was supplied regarding data availability:

The raw data has been supplied as a Supplementary File.

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
