# Peer review of "The age of onset of substance use is related to the coping strategies to deal with treatment in men with substance use disorder"

_PeerJ, doi:10.7717/peerj.3660_

## Round 0.1 · original submission · Major Revisions

· Academic Editor

Major Revisions

Thank you for your submission. Please address the concerns of the reviews, in particular the issue of shared variance identified by reviewer 1 and adding further detail to the methods section regarding recruitment.

Reviewer 1 ·

Basic reporting

This paper examines the association between age of onset of substance use and coping strategies used to deal with their treatment in patients three months abstinent. The paper is reasonably clear and focussed.



Typo line 275 ‘o’

Experimental design

I am puzzled — unless I am reading this incorrectly — why the coping strategies measured were not also administered at the beginning of treatment as presumably during treatment the participants would have been exposed, taught, and encouraged to practice a number of different coping mechanisms? It is therefore difficult to know how much coping responses were influenced by the treatment or past experience.

In addition, some explanation for why three months was chosen as surely, there are different periods within the treatment programme when skills are learnt and utilised.

There was no measure of stress so it is difficult to gage how much actual stress was generated by the treatment. If participants didn’t feel particularly stressful they wouldn’t need to apply coping strategies.

Moreover, the coping measure requested information about the ‘frequency with which’ participants used various strategies to deal with their SUD treatment. However, frequency may not be a good measure of coping as frequency does not necessary imply efficacy. Low frequency may occur for a number of reasons. One may use an approach once successfully so it would appear to have a low frequency, or an individual may use it once and not find it successful so abandon the strategy.

Some of the coping items may be influenced by the situation or the environment people are in. For example, in a residential setting if may be difficult to find someone who is a good listener, or it may not be possible to talk to someone who is close to you.

Although internal consistency for the primary scales where adequate, did the authors consider running a factor analysis on the coping items prior to analysis as there may be differences within this sample?

I didn’t see any measure of affect or mood. Mood is known to be associated with coping and coping responses?

In the regression analysis is there an issue with shared variance in the measures as the secondary measures comprise of the primary ones that are controlled earlier in the analysis?

Validity of the findings

See comments above

Additional comments

On the whole, I think this is a somewhat interesting paper. I have a few issues mainly regarding clarity that should be addressed.

·

Basic reporting

The manuscript "The age of onset of substance use influences the 1 coping strategies to deal with treatment in men with substance use disorder" is very well write and the topic is clinically and socially relevant. The results could be significant to be applied on clinical social environment.

I have few suggestions to be consider by the Authors and some comments:
- The references are relevant e sufficient, however I have a suggest on pag 9, line 96 to 98, the sentence ..."it would be which to some extent would vary depending on situation-specific variables" please review the meaning or may be add an example to clarify what they wanted to say like: which situations, which variables...

- Please consider add the statistics analyses (Chi Square, Student's t-test or Man Whitney) used on tables (1-3), as well as the significance of the statistical tests results. Also remove on of the percentage of one binary data, keeping yes or no.

- On pag 27, change "Table" by "Figure" Mean of percentile scores for the Coping Strategies Inventory. Consider removing total sample data column and keeping the cut off point for OSU≤16, OSU≥17. The figure appears unconfigured in my version.

The title "The age of onset of substance use influences the coping strategies to deal with treatment in men with substance use disorder" would be reviewed to reflect the hypothesis and/or main results. I mean, the original title seems that the age of onset of substance is cause and coping is the consequence of coping. Whit the methodological design used it is not possible to consider that it is cause and consequence.

I suggest consider reviewing the title. It could be more specific using the outcomes that really are relevant or refining more discriminant parameters obtained in the study.

Experimental design

Experimental design

Please, add more information regarding to the patients selections and where they come from, (if they come from ambulatory drug treating, in patient hospital, psychiatry clinic) as well as inviting strategy used and the interview.

On the multiple linear regression for the Coping Estrategies Inventory, I think that age should be removed on the analises. If I'm not wrong, that is regarding to the current age that the experimento was conducted (year 35.9 total sample), and the hipothesis is relates to whether they initiated substance use (OSU<=16 and OSU>=17).

Validity of the findings

I have two suggestions:

One is related to the homogeneity of time pasting since the OSU to the present time (date collect) of coping strategies. Have the authors consider that the coping strategies is not the same now (the moment that the interview was done) as use to say when they start to use drugs? Please consider discuss it at the limitation section as a confounding factor. Please, consider to discussing also if there are other potential factors that may explain the results.

The second, as the authors are working with coping strategies, I thought that one missing point is there if the emotional support receiving from the parents and social support from the family and the community, education place, and so on received during the development phase of the period when they start to use drugs. Considering that the emotional and social support is an important factor to protect against drug consumption. Another point is the life changes, as a consequence coping strategies changing as well, so the coping strategies are not the same when they start to use drugs. Is there any possible to influence of that on the results? Please add some comment.

Additional comments

The paper is very well write and the results could be significant to be applied on clinical social environment.

---

## Round 0.2 · accepted · Accept

· Academic Editor

Accept

Thank you for your re-submission. Based on the reviewers recommendation I am happy to accept your paper for publication,

Reviewer 1 ·

Basic reporting

no comment

Experimental design

no comment

Validity of the findings

no comment

Additional comments

You have clearly addressed the issues raised by me and I am satisfied with your responses. I look forward to seeing the paper when it comes out.

One minor point - perhaps 'associated' instead of 'related' in the title